# Towards Reasonable Budget Allocation in Untargeted Graph Structure Attacks via Gradient Debias

**Zihan Liu** [1,2], **Yun Luo** [1,2], **Lirong Wu** [1,2], **Zicheng Liu** [1,2], **and Stan Z. Li** [1†]
[1] AI Division, School of Engineering, Westlake University, Hangzhou, 310030
[2] College of Computer Science and Technology, Zhejiang University, Hangzhou, 310058
{liuzihan,luoyun,wulirong,liuzicheng,stan.zq.li}@westlake.edu.cn

## Abstract

It has become cognitive inertia to employ cross-entropy loss function in classification related tasks. In the untargeted attacks on graph structure, the gradients derived from the attack objective are the attacker's basis for evaluating a perturbation scheme. Previous methods use negative cross-entropy loss as the attack objective in attacking node-level classification models. However, the suitability of the cross-entropy function for constructing the untargeted attack objective has yet been discussed in previous works. This paper argues about the previous unreasonable attack objective from the perspective of budget allocation. We demonstrate theoretically and empirically that negative cross-entropy tends to produce more significant gradients from nodes with lower confidence in the labeled classes, even if the predicted classes of these nodes have been misled. To free up these inefficient attack budgets, we propose a simple attack model for untargeted attacks on graph structure based on a novel attack objective which generates unweighted gradients on graph structures that are not affected by the node confidence. By conducting experiments in gray-box poisoning attack scenarios, we demonstrate that a reasonable budget allocation can significantly improve the effectiveness of gradient-based edge perturbations without any extra hyper-parameter.

## 1 Introduction

Graph-structured data are widely used in real-world domains such as social networks [31], traffic networks [21] and recommendation systems [26]. As graphs have attracted considerable attention in fundamental researches and various applications [24, 25, 32, 13], the robustness of graph neural networks (GNNs) is rapidly gaining research attention [29]. As a major approach to robustness research, researchers have investigated various attack scenarios [19, 28, 33, 34]. The attack scenarios are classified into white-box, gray-box and black-box attacks mainly based on the information available to the attacker [27]. Unlike white-box and black-box attacks, in gray-box attacks, the graph and training labels are visible, while the network architecture of the victim model is invisible. A gray-box attacker typically transfers the attack on a trained surrogate model (i.e., an alternative GNN) to the victim model [9, 33, 34]. Therefore, gray-box attacks are an ideal scenario to study the transferability of attacks.

This paper investigates the impact of attacker's graph edge perturbation on the robustness of GNNs in the case of poisoning attacks. Poisoning attacks study the impact of the attacked data on the model training, which is also the case that may be encountered in attacks where the attacker has contaminated the data before the model training. Among the proposed gray-box untargeted attack methods, gradient-based attackers have been shown to have superior attack performance and have

---

[†]Corresponding author: Stan Z. Li.

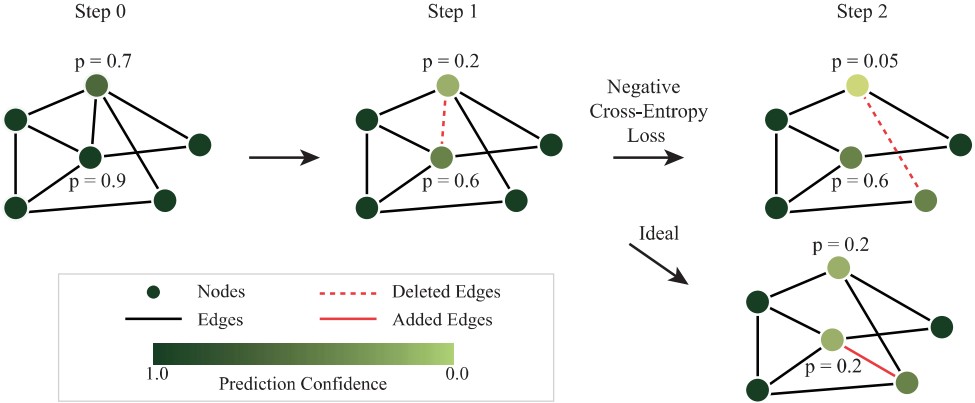

Figure 1: A schematic showing the allocation of the budget during an attack when the attack loss is the negative cross-entropy loss (i.e. $\mathcal{L}_{atk} = -\mathcal{L}_{ce}$).

become one of the mainstream attack strategies [9, 34]. Attackers are built on constructing a surrogate model, designing an attack objective function, and obtaining the gradient distribution of the adjacent matrix by backpropagation. Thereafter, the attacker selects the edges to be perturbed based on the saliency of their gradients relative to their connectivity states. We refer to the gradient distribution generated by a single node on the graph structure as the partial gradient matrix of that node. The derivation of this partial gradient matrix typically involves the mapping function of the surrogate model as well as the attack objective.

The key issue of attacks on graph structure is the structural gradients, which (along with the attribute gradients) are generated by the nodes' attack objective. The attack objective is designed as a negative cross-entropy loss in previous works [9, 34]. We define 'node confidence' specifically as the confidence of a node in the prediction of the pseudo-label class obtained from the unperturbed graph. Note that the structural gradients are generated by the loss of nodes with the message passing between nodes. We argue that there is a strong correlation between structural gradients and nodes' confidence, and we empirically prove this conjecture (demonstrated in Section 5.3). When the attack objective is negative cross-entropy loss, the partial gradient matrix from a low-confidence node has a relatively large L2 norm. Thus, low confidence nodes contribute more significant gradients to the edges; in other words, an edge is more likely to be perturbed because a low-confidence node contributes a high gradient on it. This phenomenon is defined as the gradient bias with respect to the node confidence.

Figure 1 illustrates the impact of this phenomenon in an example of a binary classification task. During the process from the initial step to the 1$^{st}$ step, the attacker reduces the confidence of a node from 0.7 to 0.2 by removing an edge, while the confidence of another node connected by this edge is reduced to 0.6. Considering that the node with a confidence of 0.2 has produced a false prediction, it is optimal for the attacker to subsequently attack the node with a confidence of 0.6. However, since the cross-entropy loss of the node with confidence 0.2 produces a large gradient, the attacker is likely to continue attacking the edges of the node based on the importance of the gradient. The reason for this phenomenon is that the cross-entropy function is a logarithm-based function. It generates higher gradients for low confidence nodes in the classification task, thus favoring the model to fit these nodes. When the attack objective is associated with the cross-entropy function, it generates more significant gradients on low confidence nodes. Since the structural gradient originates from the aggregation of node features, the effect of node confidence is reflected in the structural gradients as well.

In this paper, we demonstrate theoretically and empirically that the previous attack objective (i.e., negative cross-entropy loss) traps the attacker in a vicious cycle of inefficient budget consumption. The iterative perturbations cause the confidence level of the low confidence nodes to become lower, and the more significant gradients generated by these nodes result in them being allocated more attack budgets. To address this design flaw, this paper proposes an improved attack objective, known as gradient debias, which generates unweighted gradients on the graph structure independent of node confidence. Our approach introduces an additional weight to eliminate the bias due to the confidence-related weight on partial gradient matrices. We implement our proposed attack objective to a simple GNN surrogate model with a basic perturbation strategy to construct our attack model with Gradient Debias (GraD). To validate the effectiveness of GraD on untargeted attacks on graph structure, we

test our approach on multiple datasets as well as victim models and compare its performance with state-of-the-art baselines. The contribution of this paper can be summarized as follows:

- We demonstrate that the attack objective in previous works lead to a tendency for low-confidence nodes to contribute more significant gradients on the graph structure. This traps the attacker in a vicious cycle of constantly attacking low confidence nodes.
- We propose an improved attack objective, known as gradient debias, which solves the problem of low confidence nodes dominating the attacker's decision.
- We propose a novel attack model, GraD, which assembles our proposed attack objective and a simple surrogate model as well as a basic perturbation strategy.
- We verify the effectiveness of GraD by comparing ours' attack performance with various baselines on untargeted gray-box poisoning attacks.

## 2  Related Works

***Background*** Adversarial attacks in deep learning have received interests in various fields [30, 6, 10, 8]. The domain of adversarial attack on graph is classified in terms of scenarios of attack tasks. Attacks on graph networks contain two types of attack objectives: targeted and untargeted attacks [27]. Targeted attacks [33] disrupt the network by attacking the neighbors and edges of the target node for a given budget, while untargeted attacks try to influence the predictions of as many nodes as possible [14, 28, 34]. Depending on the knowledge of the attacker, attacks are classified as white-box [23, 28], gray-box [18, 33, 34] and black-box attacks [3, 15]. White-box attacks can obtain all the information about the victim models; gray-box attacks can obtain samples for training the victim models; black-box attacks can query the predictions of the victim models [27]. The attacker performs a poisoning attack [18, 34] if the victim model is retrained after the original data is contaminated; otherwise, it performs an evasion attack [23, 28]. In terms of the attack techniques, attacks can be classified as: modifying node attributes [14, 23], edge perturbation [9, 28, 34], and node injection [18, 19]. Each attack technique has a limited budget, such as the matrix norm variation on attributes or the changes in the adjacent matrix.

***Untargeted poisoning attack*** The researches closely related to this paper are devoted to the study of edge perturbation based untargeted attack methods. Zinger et al. [34] proposed the attack model Meta-Self, which modifies the graph structure or node features by the meta-gradient of a trained surrogate model in a gray-box setting, which is the first gradient-based attack model. Subsequently, papers [2, 9, 14, 28, 11] have also proposed new attack strategies and improvements using the gradient information provided by the surrogate model in the node features and graph structure. These works mainly use an attack objective function to obtain the gradient on the adjacency matrix by backpropagation. The form of the proposed attack objective function has been a negative cross-entropy loss. Liu et al. [12] discuss the design of the surrogate model, and the attack losses involved in graph adversarial attacks are discussed in [4]. In existing white- and gray-box attacks for classification, the form of cross-entropy is an essential component of the attack loss [28, 23, 34]. Our proposed confidence-oriented attack loss overturns this mindset, which has been justified in this paper.

## 3  Preliminaries

### 3.1  Notations

For the node classification task, given a graph $G$ with node attributes $X$ and a subset of node labels $Y$, GNNs are expected to predict the class of unlabelled nodes. Let $G = (V, E, X)$ represents an attribute graph, where $V = \{v_1, v_2, ..., v_n\}$ is the set of $N$ nodes, $E \subseteq V \times V$ is the set of edges. Each sample is represented as a node $v_i$ associated with its attribute $x_i \in \mathbb{R}^d$ and label $y_i \in \mathbb{R}^k$, where $k$ is the number of classes. We denote the adjacency of the nodes as a binary adjacent matrix $A = \{0, 1\}^{N \times N}$, where $A_{i,j} = 1$ if $(i, j) \subseteq E$. In addition, the surrogate model is denoted by $f_\theta$, while the prediction of $f_\theta$ for node $v_i$ is denoted by the probability distribution $P_{v_i}$. We use $\mathcal{L}$ to represent the loss (objective) function. The perturbed graph is indicated as $G' = (A', X')$ to distinguish the perturbed graph from the original graph. The attack on a graph is limited by a budget $\Delta$.

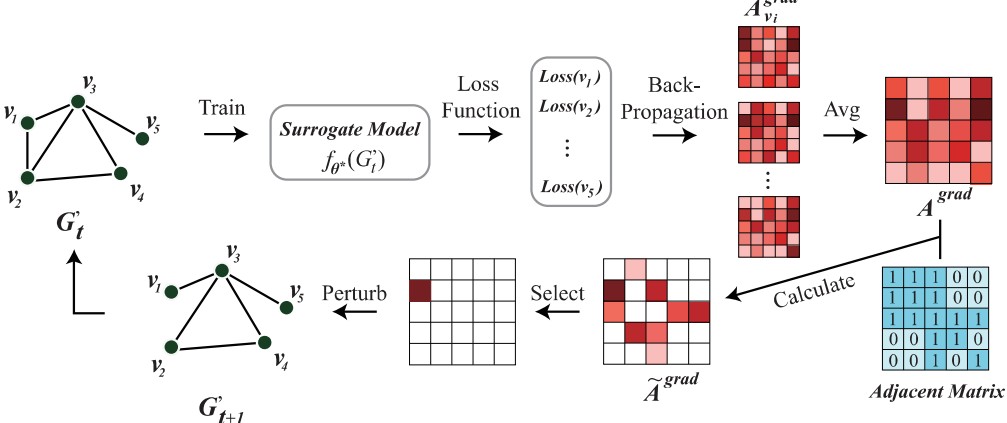

Figure 2: Illustration of the process by which the attack model generates one perturbation.

## 3.2 Gradient-based Attack on Graph Structure

In scenarios where the attacker performs untargeted attacks on graph structure, the attacker is restricted by attack imperceptibility. The $l_0$ norm of the difference in the perturbed graph with respect to the original graph is bounded by the perturbation budget $\Delta$. For an undirected graph, $\Delta$ is expressed as:

$$\|A - A'\|_0 \leq 2\Delta. \tag{1}$$

$\Delta$ is generally no more than 5% of the number of edges in the original graph. In an untargeted attack scenario, the attacker aims to reduce the test performance of the poisoned victim model.

Gradient-based attackers have become mainstream approaches of edge perturbations on the graph structure [9, 28, 34]. Unlike the gradient-based attacks widely used in computer vision [29], for graph data, the discretization of the graph structure dictates that the gradient matrix cannot be directly added to the adjacency matrix. The gradient on the graph structure $A^{grad}$ is considered as a reference for selecting edge candidates that are more likely to be ideal for perturbations. It can be derived by the following equations:

$$\theta^* = \underset{\theta}{\arg\min} \, \mathcal{L}_{ce}(f_\theta(G), Y), \tag{2}$$

$$A^{grad} = \nabla_A \mathcal{L}_{atk}(f_{\theta^*}(G)), \tag{3}$$

where $\mathcal{L}_{atk}$ is the attack loss function and $f_{\theta^*}$ is the properly trained surrogate model. In existing methods, the attack loss is expressed as

$$\mathcal{L}_{atk} = -\sum_{i=1}^{V^*} \mathcal{L}_{ce}(f_{\theta^*}(G), y_i), \tag{4}$$

where $V^*$ is a subnet of nodes that involved in calculating the attack loss. The subset $V^*$ consists of two forms: $V^*_{train}$ and $V^*_{self}$, where $V^*_{train}$ contains labeled nodes and $V^*_{self}$ uses pseudo-labels generated by the surrogate model so it can contain unlabelled nodes. For the edge between nodes $v_i$ and $v_j$, if $A_{i,j} = 1$ and $A_{i,j}^{grad} < 0$, or if $A_{i,j} = 0$ and $A_{i,j}^{grad} > 0$, then flipping edge $E_{i,j}$ is considered as a perturbation that has the potential to negatively affect the victim model. The process of perturbing the graph using the gradient information can be represented as:

$$A'_t = \phi(\nabla_A \mathcal{L}_{atk}(f_{\theta^*}(G'_{t-1})), A'_{t-1}), \tag{5}$$

where $\phi$ denotes the strategy for choosing the edge to be attacked. The factors that influence the perturbation include the attack loss $\mathcal{L}_{atk}$ as well as the strategy $\phi$ and the surrogate model $f_{\theta^*}$.

In the work [34], the authors do an ablation study on the node sets $V^*_{train}$ and $V^*_{self}$. Their results show that constructing the attack loss using $V^*_{self}$ leads to superior performance. This is due to the fact that node classification tasks are generally semi-supervised and therefore the labeled nodes are too sparse respect to the whole graph. However, we find a fatal problem during the exploration of the attack loss. The cross-entropy is a function related to the prediction confidence of the node on the

labeled class $P_{v_i}(y_i|f_{\theta^*}(G))$. That is, the gradient from the cross-entropy function is also correlated with the confidence. The result of this correlation is that the gradient generated by the different nodes will contain a magnitude. In the next section, we will discuss the reason why this magnitude appears and whether it is reasonable in the process of attack.

## 4  Methodology

### 4.1  The Pipeline of Attack Model

We first present the overview of our attack model in Figure 2, which illustrates the entire flow of an attacker perturbing an edge of a graph.

The first step is to retrain a surrogate model $f_{\theta^*}(G'_t)$ with the perturbation graph $G'_t$ in the way of Eq. (2). We use the surrogate model with same network architecture as [34], which is a linear 2-layer graph convolutional network (GCN) [7]: $f_\theta(G) = \text{softmax}(\hat{A}^2 X W)$, where $\hat{A} = D^{-1/2}(A+I)D^{-1/2}$ is the normalized adjacent matrix and $W$ is a learnable weight matrix.

In the second step, we use the pseudo-label to construct the attack loss of the node. The pseudo-label $y'_i$ is calculated from the prediction of the surrogate model $f_{\theta^*}(G)$ trained with the original graph $G$. Note that we set the attacker's surrogate model to be consistent with the model used to generate pseudo-labels to avoid inconsistent predictions due to different structures of GNNs. The attack loss $\mathcal{L}_{atk}$ of node $v_i$ is related to the confidence $P_{v_i}(y'_i|f_{\theta^*}(G))$ and the pseudo-label $y'_i$, the details of which will be discussed in Section 4.2.

In the third step, the attack loss of each node is backpropagated to produce a partial gradient matrix $A^{grad}_{v_i}$ according to Eq. (3). A partial gradient matrix $A^{grad}_{v_i}$ denotes the structural gradient generated from a single node $v_i$. For a GCN that aggregates $h$ times, a node will generate gradient values on all edges between that node and its $h$-hop neighbors. The overall gradient information $A^{grad}$ passed to the attack strategy $\phi$ is the average of all the partial gradient matrices, which can be represented as:

$$A^{grad} = \frac{1}{N^*}\left(\sum_{v_i}^{V^*} A^{grad}_{v_i}\right), \tag{6}$$

where $N^*$ denotes the number of nodes in the node subset $V^*$ which is chosen as the set of unlabelled nodes in our implementation. The attack strategy $\phi$ is a basic edge perturbation selection method based on the saliency of gradients, which is represented as follows:

$$\widetilde{A}^{grad} = \begin{cases} A^{grad}_{i,j} & \text{if } A^{grad}_{i,j} > 0 \ \& \ A_{i,j} = 0 \\ -A^{grad}_{i,j} & \text{if } A^{grad}_{i,j} < 0 \ \& \ A_{i,j} = 1 \\ 0 & \text{else} \end{cases} \tag{7}$$

where $\widetilde{A}^{grad}$ in Eq. (7) removes candidates with inconsistent gradient signs and edge states. Then the attacker flips the edge $e'_{i,j} = \text{argmax}_{(i,j)} \widetilde{A}^{grad}$, with maximum gradient saliency as the $t$-th perturbation, which forms the perturbed graph $G'_{t+1}$ that runs into the next loop.

### 4.2  Budget Allocation with Gradient Debias

After explaining the attack pipeline, we explain why low confidence nodes dominate the attack because of the previous attack objective, i.e., negative cross-entropy loss $-\mathcal{L}_{ce}$. We first merge the formulation Eq. (3) and Eq. (6):

$$A^{grad} = \sum_{v_i}^{V^*} \nabla_A \mathcal{L}_{atk}(P_{v_i}, y'_i), \tag{8}$$

The gradient on the graph structure $A^{grad}$ comes from the contribution of each node in $V^*$. The widely used formulation for attack objective is the negative cross-entropy loss, shown in Eq. (4). The cross entropy loss $\mathcal{L}_{ce}$ can be expressed as:

$$\mathcal{L}_{ce}(f_{\theta^*}(G), y'_i) = -\sum_k y_k \log P_{v_i}(y_k|f_{\theta^*}) = -\log P_{v_i}(y'_i|f_{\theta^*}), \tag{9}$$

where the $\mathcal{L}_{ce}$ is a function of the prediction confidence of the node $v_i$ on the labeled class $y'_i$. According to the chain rule, the gradient on the adjacent matrix derived from attack loss is:

$$\nabla_A \mathcal{L}_{atk} = \frac{\partial \mathcal{L}_{atk}}{\partial P_{v_i}(y'_i|f_{\theta^*})} \cdot \frac{\partial P_{v_i}(y'_i|f_{\theta^*})}{\partial A}, \tag{10}$$

where term $\partial P_{v_i}(y'_i|f_{\theta^*})/\partial A$ can produce a matrix of gradient distributions influenced by the mapping function of surrogate model, and term $\partial \mathcal{L}_{atk}/\partial P_{v_i}(y'_i|f_{\theta^*})$ acts as a weight to it. Next, we expand this expression for this weight:

$$\frac{\partial \mathcal{L}_{atk}}{\partial P_{v_i}(y'_i|f_{\theta^*})} = \frac{-\partial \mathcal{L}_{ce}}{\partial P_{v_i}(y'_i|f_{\theta^*})} \quad = P_{v_i}(y'_i|f_{\theta^*})^{-1}. \tag{11}$$

From the Eq. (11), we observe that the magnitude of a partial gradient matrix is weighted by the reciprocal of node confidence $P_{v_i}(y'_i|f_{\theta^*})^{-1}$.

When the attack loss is cross-entropy-based, nodes with low confidence generate gradients with a larger magnitude than those that nodes with high confidence generate. For example, for a node with a confidence of 0.1, the derivative of its attack loss with respect to confidence is 10, which is four times larger than that of a node with a confidence of 0.4. Since the selection of perturbations depends on the significance of gradients, nodes with low confidence are more likely to contribute more significant gradients in their partial gradient matrices. This causes a low confidence node to become more vulnerable when its confidence is reduced.

Note that the budget $\Delta$ should be allocated through the entire graph. It is ineffective to allocate budget to a node whose confidence is low enough in the labeled class. The previous attack loss results in a tendency; that is, nodes with lower confidence are more likely to influence the judgment of the attacker. Therefore, with the increase of low confidence nodes during the attack process, it becomes more difficult for the attacker to allocate the budget to other nodes, resulting in budget waste.

To address the above problem, we propose a simple but effective attack objective known as gradient debias. Since the attack objective based on cross-entropy is what weights the partial gradient matrices, a solution is to apply an additional weight $\lambda_{P_i}$ to balance the former weight. We propose our gradient debias attack objective, of which the expression is:

$$\lambda_{P_i} = P_{v_i}(y'_i|f_{\theta^*}) \tag{12}$$

$$\mathcal{L}_{atk-gd} = \lambda_{P_i}(-\mathcal{L}_{ce}(f_{\theta^*}(G), y'_i)) = \lambda_{P_i} \log P_{v_i}(y'_i|f_{\theta^*}), \tag{13}$$

where $\lambda_{P_i}$ is a detached term that is numerically equal to the prediction confidence. Considering Eq. (10) and (11), our attack objective regularize each partial gradient matrices $A_{v_i}^{grad}$ in the overall gradient matrix $A^{grad}$. Since the effect of $\lambda_{P_i}$ essentially cancels out the effect of $\partial \mathcal{L}_{atk}/\partial P_{v_i}(y'_i|f_{\theta^*})$ in Eq. (10), Eq. (13) can be rewrited to a simple form:

$$\nabla_A \mathcal{L}_{atk-gd} = \frac{\partial P_{v_i}(y'_i|f_{\theta^*})}{\partial A} \tag{14}$$

$$\mathcal{L}_{atk-gd} = P_{v_i}(y'_i|f_{\theta^*}). \tag{15}$$

Compared with $\mathcal{L}_{atk}$, $\mathcal{L}_{atk-gd}$ has a simpler formula, which can be understood as globally reducing the confidence of the nodes. Based on the gradient debias, the importance of all nodes during the attack is equal for the attacker. In the previous $\mathcal{L}_{atk}$, if there is a low confidence node affecting the gradient of this edge (i.e., node $v_i$, $v_j$ or their $(h-1)$-hop neighbors), then the contribution of this node to the gradient of edge $A_{i,j}$ will be significant. It explains why the previous approach is more likely to waste the attack budget on low confidence nodes. In our proposed $\mathcal{L}_{atk-gd}$, the gradients generated by each node are equally superimposed on edges. That is, the gradient on edge $A_{ij}^{grad}$ reflects the expected confidence decrease of perturbing this edge, regardless of the confidence of the nodes. During the perturbation, $\mathcal{L}_{atk-gd}$ prevents the attacker from falling into the vicious circle of repeatedly making low confidence nodes even lower and gives the attacker more reliable gradient information leading to a more reasonable budget allocation.

Table 1: Experimental results of 'weak transfer' scenarios, where the victim models are GCNs. 'Clean' denotes the result for an unperturbed graph. The best results from each experiment are bold. The Gain row indicates the improvement of our method relative to the second-best method.

| | Cora | | Cora-ML | | Citeseer | | Polblogs | |
|---|---|---|---|---|---|---|---|---|
| Pert Rate | 3% | 5% | 3% | 5% | 3% | 5% | 3% | 5% |
| Clean | 81.7±0.3 | | 84.0±0.4 | | 69.9±0.4 | | 95.4±0.4 | |
| Random | 81.7±0.3 | 81.2±0.3 | 83.0±0.5 | 82.8±0.4 | 69.0±0.3 | 67.8±0.4 | 85.1±0.6 | 84.0±0.9 |
| DICE | 81.2±0.3 | 80.9±0.5 | 83.5±0.5 | 82.5±0.4 | 69.3±0.2 | 69.0±0.4 | 81.9±0.2 | 79.2±0.7 |
| EpoAtk | 79.2±0.7 | 77.0±0.6 | 82.1±0.3 | 81.3±0.4 | 67.3±0.5 | 66.3±0.4 | 83.1±1.1 | 82.6±0.3 |
| Meta-Train | 79.5±0.4 | 76.4±0.2 | 81.7±0.3 | 79.0±0.3 | 67.0±0.4 | 65.5±0.4 | 87.7±0.3 | 83.5±0.7 |
| Meta-Self | 77.9±0.4 | 75.8±0.4 | 79.5±0.3 | 76.2±0.3 | 65.0±0.3 | 60.4±0.4 | 79.4±0.5 | 75.8±0.5 |
| **GraD(ours)** | **75.2±0.4** | **71.4±0.5** | **77.9±0.1** | **75.0±0.3** | **61.2±0.7** | **55.8±0.9** | **74.4±0.3** | **69.5±0.4** |
| Gain | +2.7 | +4.4 | +1.6 | +1.2 | +3.8 | +4.6 | +5.0 | +6.3 |

# 5 Experiments

In this section, we conduct experiments to verify the effectiveness of our proposed attack model, GraD [1]. We describe the experimental scenarios, including the datasets, baselines, and the experiment setup. The practical tests of the computational efficiency of GraD and other attack models are provided in Appendix A.2.

## 5.1 Experimental Scenarios

***Datasets*** We adopt datasets of Citeseer [17], Cora [16], Cora-ML [16], and Polblogs [1] to evaluate our approach, which are detailed in Appendix A.1. Following the setup in [34], datasets are split to 10% of labeled nodes and 90% of unlabeled nodes. The ground-truth labels of unlabeled nodes are invisible to both the attacker and the surrogate and are only used to evaluate the performance of the adversarial attacks. For the integrity of the experiments, we also provide a comparison on the largest connected component (LCC) datasets in Appendix A.3.

***Baselines*** This paper takes Random, DICE [22], Meta-Train[34], Meta-Self [34], EpoAtk[9] (a white-box attack model in the original paper transferred to gray-box attack) as baselines, all of which attack GNNs by modifying graph structure. All baselines are open source, which makes them easy to replicate. The implementation of baselines is detailed as follows.

***Experimental Setup*** We conduct experiments in untargeted gray-box poisoning attack scenarios. The gray-box attack scenarios are further split into weak transfer and strong transfer. In a weak transfer scenario, the victim model is also a GCN [7] as the surrogate model but has different network structures. In a strong transfer scenario, the victim model has different principles from the surrogate, which is a GAT [20] or a GraphSage [5]. Different GNNs tend to differ from graph representations, challenging the attack transferability. To ensure absolute fairness in the test scenarios, we save the perturbation graphs generated by all attack models and test them on unified victim models. Each experiment is repeated ten times with different initialization, and we take the mean and variance as a result. We compare the attack performance of each method at perturbation rates of 3% and 5%. The results of the experiment are expressed in terms of classification accuracy.

## 5.2 Performance of Attack Model

Experiments are designed to compare our proposed GraD with several baselines, and the experimental results are presented in this section. We category the experiments into weak transfer scenarios and strong transfer scenarios based on the differences between the surrogate and victim models. In gray-box attack, experimental scenarios are designed to verify the transferability of the attack methods. In a weak transfer scenario, both the victim model and the surrogate model are GCNs. Although they differ in the design of activation functions and network layers, their similar representation learning principles make the attack in this scenario relatively simple. In a strong transfer scenario, the victim model is a GAT or a GraphSage, unlike the surrogate model. They are different in principle from representation learning, making attacks in this scenario more difficult. The attacking budget $\Delta$ is set at 3%, and 5% of edges in the original graph in the weak transfer scenarios, while it is set at 5% in the strong transfer scenarios. The control group 'Clean' represents the classification accuracy of the victim model trained with the original graphs under the same training process.

---
[1] https://github.com/Zihan-Liu-00/GraD

Table 2: Experimental results of 'strong transfer' scenarios, where GAT and GraphSage act as victim models. The experiments are all performed at a perturbation rate of 5%.

| Victim | Cora | | Cora-ML | | Citeseer | | Polblogs | |
|---|---|---|---|---|---|---|---|---|
| | GAT | GraphSage | GAT | GraphSage | GAT | GraphSage | GAT | GraphSage |
| Clean | 81.4±0.6 | 80.8±0.4 | 82.3±0.3 | 81.9±0.6 | 68.2±0.6 | 69.8±0.5 | 95.0±0.3 | 90.5±4.4 |
| Random | 80.3±0.6 | 80.7±0.6 | 80.7±0.6 | 80.6±0.8 | 66.0±0.7 | 68.7±0.7 | 83.7±0.3 | 78.1±1.1 |
| DICE | 79.6±0.8 | 80.0±0.4 | 80.9±0.5 | 80.7±0.9 | 66.5±0.7 | 69.1±0.8 | 81.8±0.2 | 76.1±1.9 |
| EpoAtk | 79.4±0.7 | 78.9±0.8 | 79.4±0.6 | 80.2±0.7 | 66.9±0.6 | 68.8±0.5 | 85.6±0.7 | 69.4±1.8 |
| Meta-Train | 78.5±0.6 | 76.1±1.0 | 77.2±0.7 | 77.7±1.8 | 66.5±0.6 | 67.9±0.9 | 86.0±0.8 | 71.2±4.0 |
| Meta-Self | 77.5±0.4 | 74.9±0.8 | 74.2±0.5 | 76.3±1.6 | 60.3±0.8 | 60.8±0.6 | 83.4±1.6 | **64.1±2.6** |
| **GraD(ours)** | **74.1±0.4** | **70.6±1.1** | **72.9±0.6** | **74.6±1.0** | **60.1±0.8** | **58.8±2.2** | **70.8±0.8** | 66.0±0.4 |
| Gain | **+3.4** | **+4.3** | **+1.3** | **+1.7** | **+0.2** | **+2.0** | **+11.0** | **-1.9** |

### 5.2.1 Weak Transfer Scenarios

Table 1 shows the experimental results when the network structures of the victim model and the surrogate model are GCNs. Our proposed GraD uses a linear GCN network similar to [34], while EpoAtk [9] is a nonlinear GCN network with a different structure. In this scenario, the representation learning process of the surrogate model and the victim model is similar, so the transfer of the attack is less difficult. From the table 1 we can see that GraD is superior to the existing state-of-the-art baselines in all untargeted poisoning attack experiments and considerably outperforms other work in most scenarios. In terms of overall attack performance, GraD is far better than the other attack models, followed in order by Meta-Self, Meta-Train, and EpoAtk, while Random and DICE, which are based on randomness, are the worst of all baselines. GraD outperforms the second-best method on the Cora dataset by 2.7% and 4.4%. On Cora-ML, GraD performs better than the second-best method by 1.6% and 1.2%. On Citeseer, GraD obtains better results than the second-best method by 3.8% and 4.6%. On Polblogs, GraD outperforms the second-best method by 5.0% and 6.3%. Our method was **on average 4.48%** higher than the second-best method Meta-Self, with the best results being 1.2% to 8.0% higher than the second-best. From the results shown in Table 1, the performance of GraD exceeds that of other state-of-the-art baselines, which proves that our proposed attack model substantially improves the performance of the attacker. We experimentally validate our discussion of design flaws on attack loss function in prior work and the effectiveness of our proposed gradient de-weighting.

### 5.2.2 Strong Transfer Scenarios

Table 2 shows the experimental results for the case where the structures of the victim model and the surrogate model are different. The surrogate model is a GCN model, while the victim model is a GAT or a GraphSage. Results show the performance of GraD and baselines at perturbation rates of 5%. Comparing the results in Table 2 with that in Table 1, although both scenarios belong to the gray-box setting, the attack method yields a more significant deviation when the structure of the victim model differs from the surrogate. On Cora, Cora-ML, and Citeseer, there is consistency between the effects of each attack method in the strong transfer scenarios and those in the weak transfer scenarios. For the scenarios where the victim models are GATs, our method improves 3.4%, 1.3%, and 0.2% compared to the second-best method on Cora, Cora-ML, and Citeseer, respectively. For the scenario where the victim model is GraphSage, our method compared to the second-best method improves by 4.3%, 1.7%, and 2.0% on the three datasets. The performance of each method on Polblogs varies. When the victim model is GAT, our method outperforms the second best method, DICE, by 11%, while the random-based DICE outperforms other gradient-based baselines. However, when the victim model is GraphSage, the effect of GraD is 1.9% lower than the best model. This result shows that the variability of the network structure is what makes the transfer of attacks more difficult. Overall, GraD still substantially improves attackers' performance in strong transfer scenarios, demonstrating the effectiveness of our approach.

### 5.3 Correlation between Nodes' Confidence and Structural Gradients

An important assumption supporting our theory is that the confidence degree of nodes is strongly correlated with the gradient of the edge. To verify our statement that lower confidence nodes produce relatively significant gradients, we do the statistics shown in Figure 3. It compares the previous attack objective $-\mathcal{L}_{ce}$ with $\mathcal{L}_{atk-gd}$ on Citeseer and Cora. In the figure, the x-axis of each point is the

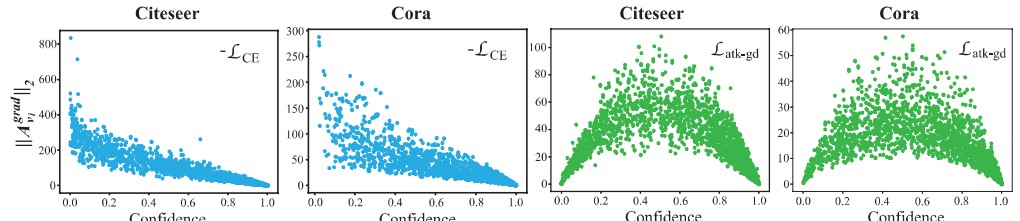

Figure 3: Scatterplot of the nodes' confidence with the $\ell_2$ norm of their partial gradient matrices.

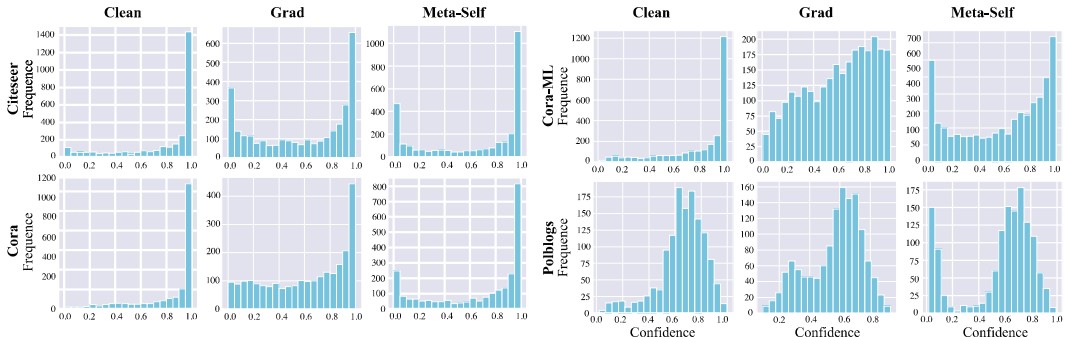

Figure 4: Confidence distribution of perturbed graphs from GraD and Meta-Self on a unified GCN model. The x-axis is the interval of the confidence level, and the y-axis is the frequency of the node confidences distributed in the interval.

confidence of the node on the pseudo-label class, and the y-axis is the $\ell_2$ norm of its partial gradient matrix (i.e., the global salience of the gradient generated by that node via the attack loss).

The previous attack objective produces significant gradients for low confidence nodes. The attack is more susceptible to low confidence nodes, which limits budgetary planning to a local scope. The gradient bias of low confidence nodes is eliminated in our proposed $\mathcal{L}_{atk-gd}$. The gradient is naturally larger on a node at a middle confidence level than at a low or high confidence level. This is because the node deviates from the centroid of any class in the hidden space. Nodes at the middle confidence level become low confidence nodes when attacked, while low confidence nodes are not vulnerable. It ensures that the vulnerable nodes continuously change and achieves a global budget allocation.

## 5.4 Confidence Distribution Analysis

To demonstrate that our approach solves the problem we state, we visualize the distribution of the surrogate model's prediction confidence, as shown in Figure 4. The confidence level of a node is the output possibility (obtained from the clean graph) for its pseudo-label class. We compare the confidence distributions between the perturbed graphs generated by GraD and Meta-Self. 'Clean' represents the confidence distribution of the GNN model trained with the unperturbed graph. The GNN model for each graph is set up as a GCN with unified network architecture.

In Cora, Cora-ML, and Citeseer, most nodes have confidence levels greater than 0.9 in a clean graph, mainly distributed at the high confidence level. The nodes in the Polblogs are mainly distributed at a confidence level of about 0.7. For the perturbed graphs generated by Meta-Self, the number of nodes with high and low confidence is significantly more than other nodes, especially in Cora, Citeseer, and Polblogs. It validates our elaboration that the previous attack objective leads the attacker into repeatedly attacking low confidence nodes. Comparing GraD with Meta-Self, graphs perturbed by GraD have fewer nodes being distributed in the high confidence level. Moreover, the graphs from GraD have fewer nodes on the low confidence level than that of Meta-Self. The results show that GraD attacks more high-confidence nodes and generates fewer low-confidence nodes than Meta-Self. The results illustrates that GraD allocates the attack budget on low confidence nodes to high confidence nodes that have not yet been attacked.

# 6  Conclusion

This paper focuses on the budget allocation problem for untargeted gray-box edge perturbations. We first argue that the loss function used by other methods will lead to an unreasonable budget allocation during the perturbation process. The gradient of the attack loss function based on cross-entropy is related to the confidence of the node on the pseudo-label, which leads to the problem that nodes with lower confidence are more likely to be allocated budgets. Considering that a node is successfully attacked after its confidence is below a threshold, it should not be allocated more budget than other nodes from a budget allocation perspective. To this end, we propose GraD, an attack model using a gradient de-weighting attack objective. Our method proposes a simple but effective attack objective to balance the harmful tendency caused by confidence when the gradient is generated through backpropagation. In the experimental section, we validate the effectiveness of our proposed method by experimenting with different settings of the untargeted poisoning gray-box attack. The results demonstrate that GraD significantly improves the attacker's attack success rate, which proves the reliability of our discussion on the shortcoming of the previous attack objective and the effectiveness of our approach.

## Acknowledgement

This work is supported in part by Ministry of Science and Technology of the People´s Republic of China (No. 2021YFA1301603) and National Natural Science Foundation of China (No. U21A20427).

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
