# A    Appendix

## A.1    Detail of Datasets

Table 3: Statistics of datasets.

| Datasets | Vertices | Edges | Classes | Features |
|----------|----------|-------|---------|----------|
| Citeseer | 3312 | 4732 | 6 | 3703 |
| Cora | 2708 | 5429 | 7 | 1433 |
| Cora-ML | 2995 | 8416 | 7 | 2879 |
| Polblogs | 1222 | 16714 | 2 | - |

## A.2    Computational Efficiency

We compared the computational efficiency of our method with the baseline, shown in Table 4. The table counts the running time of each method from the beginning to the completion of the perturbation process. The experiment is run at a uniform perturbation rate of 5%. Moreover, the experiments on both baselines and our approach are implemented under the environment of the PyTorch 1.3.1 library with Intel(R) Xeon(R) Gold 6240R @ 2.40GHz CPU and NVIDIA V100 GPU. DICE has the highest computational efficiency since it depends on randomness without gradient derivation. The computational efficiency of EpoAtk is the lowest because it carries an exploration module that requires additional computation. Our method is slightly more computationally efficient than Meta-Train and Meta-Self, among the other methods. Therefore, as we state, the computational efficiency of GraD is satisfactory while the performance is competitive.

Table 4: Comparison of computational efficiency.

| Methods | Cora | Cora-ML | Citeseer | Polblogs |
|---------|------|---------|----------|----------|
| DICE | 3s | 3s | 4s | 1s |
| EpoAtk | 1057s | 1771s | 705s | 1878s |
| Meta-Train | 205s | 519s | 386s | 367s |
| Meta-Self | 210s | 533s | 393s | 429s |
| Grad(ours) | 200s | 513s | 373s | 385s |

## A.3    Attack Performance in LCC Datasets

To further confirm the validity of our method, we replicate part of the experimental attack scenarios in Meta-Train & Meta-Self. The datasets Cora, Cora-ml, and Citeseer contain singleton nodes that are not connected to other nodes. These nodes are removed from the graph in these datasets, named LCC datasets. The perturbation rate of the attack scenario is set to 5%, and the victim network architecture is nonlinear two-layer GCN. The experimental results are shown in Figure 5. Comparing the classification accuracy of the attacked graphs, GraD has a 3-4% higher attack success rate than the second-ranked Meta-Self. Among other models, EpoAtk performs better than Meta-Train. In the replicated attack scenario, GraD still demonstrates strong competitiveness.

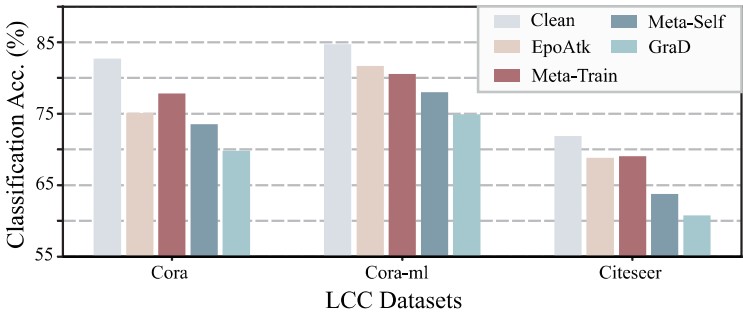

Figure 5: Performance of attack models on LCC datasets.

# B  Broader Impact

This paper proposes a poisoning attack model on graph-structured data. Our approach is able to significantly affect the robustness of graph neural networks by modifying limited edges, which puts it at risk of malicious use. The attack method proposed in this paper requires 1.attributes of nodes 2.training labels 3.graph structure. We remind the owners to protect the privacy of the data to avoid malicious attacks.