# OpenReview forum: "Towards Reasonable Budget Allocation in Untargeted Graph Structure Attacks via Gradient Debias"
_NeurIPS.cc/2022/Conference — NeurIPS 2022 Accept_

### Official Review · Reviewer_CLc8 · 2022-07-04

**Rating:** 4
**Confidence:** 4
**Soundness:** 3 good
**Presentation:** 2 fair
**Contribution:** 2 fair

**Summary:**

This paper studies the graph modification attack in a untargeted poisoning setting by editing the edges, and reveals that using the negative cross entropy as the attack loss will cause unreasonable budget allocation issue. Thus, the paper proposes a new attack objective by plugging the confidence to reduce the influence of the negative cross entropy. The authors conduct some experiments to validate their findings.

**Questions:**

Starting from an interesting observation of the budget allocation in the untargeted poisoning graph modification attack, the authors reveal that using negative cross entropy loss can affect the gradients signals passed to the adversary hence the adversary keeps allocating budgets to attack the low confidence nodes. However, there are several concerns raised when I read the paper.

1. The scope of the studied attack might be too limited. In fact, graph adversarial attacks can have both poisoning and evasion, targeted and untargeted, modification and injection attacks [1,2,3]. Especially, [3,4] point out that the poisoning attack can have severe issues due to the ill-defined imperceptibility. More justification and discussions are required for the reasons about why adopting the setting in the paper. Otherwise, the limited setting used and discussed in this paper weakens the significance and novelty of the paper, given that the main contribution of this work is to propose a new adversarial objective for graph adversarial attacks.

2. Important experiments seem to be missed in the paper. Since the proposed objective can improve the budget allocation of the graph adversarial attack, the authors only conduct two experiments with 2 specific allowed budgets, which is significantly less than existing literatures [1,2,3,4,5,6,7], providing weak support to the effectiveness of the proposed objective.

3. It would be more interesting if the authors could discuss and evaluate the budget allocation during the targeted attack to the large scale graphs [8,9] which tends to be appealing in practice, as we often do not have sufficient budgets to attack all of the test nodes in a large scale graph while many of node classification applications focus on the large scale graphs. In the literature, the graph injection attack can serve a good proxy [4].

4. The scope of the experiments is also limited. The authors only test with GCN, GAT, GraphSage, while neglecting many defense methods [6,7], as well as methods such as Nettack that can also be leveraged to attack and see the survey [1,2,3] for more details. Some representative datasets such as Pubmed are missed.

5. In experimental setup, ``Each experiment is repeated ten times with a fixed random seed’’, what is the point to fixing the random seeds and repeat the experiments? How do they produce different results if the random seeds are fixed?

6. The writing is hard to follow. There are too many undefined mathematical symbols and typos. Here are some of them (not sure why authors cancel the line numbers, making the reviewers hard to point out the specific locations of their concerns):
    - In 3.1, ``while the prediction of f θ is denoted by the probability distribution P_vi’’
    - In Eq. 4, what is z_i?
    - After Eq. 4, `` where V ∗ is a subnet of nodes’’
    - In Sec. 4, is it the linearized GNN, SGC or GCN?
    - In Sec. 4, y_i is undefined.
    - In Eq. 6, what is A^grad? What is A^grad with subscript v_i? Does it mean taking some entry in A^grad?
    - After Fig. 4, ``We consider this to be since the node deviates from the centroid of any class.’’




References:

[1] Wei Jin, Yaxin Li, Han Xu, Yiqi Wang, Shuiwang Ji, Charu Aggarwal, Jiliang Tang. Adversarial Attacks and Defenses on Graphs: A Review, A Tool and Empirical Studies. SIGKDD Explorations 2020.

[2] Lichao Sun, Yingtong Dou, Carl Yang, Ji Wang, Philip S. Yu. Adversarial Attack and Defense on Graph Data: A Survey. arXiv 2020.

[3] Qinkai Zheng, Xu Zou, Yuxiao Dong, Yukuo Cen, Da Yin, Jiarong Xu, Yang Yang, Jie Tang. Graph Robustness Benchmark: Benchmarking the Adversarial Robustness of Graph Machine Learning. NeurIPS 2021 Datasets and Benchmark Track.

[4] Yongqiang Chen, Han Yang, Yonggang Zhang, Kaili Ma, Tongliang Liu, Bo Han, James Cheng. Understanding and Improving Graph Injection Attack by Promoting Unnoticeability. ICLR 2022.

[5] Shuchang Tao, Qi Cao, Huawei Shen, Junjie Huang, Yunfan Wu, Xueqi Cheng. Single Node Injection Attack against Graph Neural Networks. CIKM 2021.

[6] Xiang Zhang, Marinka Zitnik. GNNGuard: Defending Graph Neural Networks against Adversarial Attacks. NeurIPS 2020.

[7] Wei Jin, Yao Ma, Xiaorui Liu, Xianfeng Tang, Suhang Wang, Jiliang Tang. Graph Structure Learning for Robust Graph Neural Networks. KDD 2020.

[8] Simon Geisler, Tobias Schmidt, Hakan Şirin, Daniel Zügner, Aleksandar Bojchevski, Stephan Günnemann. Robustness of Graph Neural Networks at Scale. NeurIPS 2021.

[9] Enyan Dai, Tianxiang Zhao, Huaisheng Zhu, Junjie Xu, Zhimeng Guo, Hui Liu, Jiliang Tang, Suhang Wang. A Comprehensive Survey on Trustworthy Graph Neural Networks: Privacy, Robustness, Fairness, and Explainability. arXiv 2022.


**Limitations:**

The authors did not provide such a discussion.

**Strengths And Weaknesses:**

*Originality & Significance*: Studies around graph adversarial attack and defense are of great importance to the community. However, given the limited scope of the focus in this paper, the significance might be weakened. See the Questions below for more details.

*Quality*: The authors conduct certain theoretical and empirical analysis to validate their claims. However, the experiments might not be sufficient to fully support the effectiveness of the proposed method. See the Questions below for more details.

*Clarity*: The paper is well-organized yet not easy to follow. There are too many undefined mathematical symbols and typos.

---

> ### Author Response · Authors · 2022-08-02
> **Rebuttal response by authors**
>
> We are grateful to you for your effort in reviewing our paper. We will next respond to the weaknesses and questions you mentioned. Before answering the questions, we would like to sort out and highlight the practical contribution of this work, as well as to elaborate on the points of focus on which we designed the experiment.
> ****
> **The contribution of this work:**
> The main contribution of this paper is to *break the stereotype that "attack objective is the convert of classification objective"* and propose that their objectives should be inconsistent. We theoretically and empirically analyze the existence of this problem and propose a *confidence-based* attack objective as a solution. We expect this discovery to be an inspiration for others’ researches on utilizing graph structure gradients.
> 1. **Principal contribution**: we first present the design flaw of the widely used cross-entropy attack loss from the perspective of budget allocation. This is a counter-intuitive conclusion since cross-entropy is widely used in the classification-related tasks. We give mathematics to prove that cross-entropy leads to the problem of unreasonable budget allocation.
> 2. **Secondary contribution**: we propose an improved attack objective that abandons the form of cross-entropy and instead designs loss based on the confidence of the pseudo-labeled class.
> 3. **Experimental validation**: The experiments aim to proof the existence of the budget allocation problem for CE-based attack loss and validate our proposed attack loss by comparing the attack performance. In addition, we design analytical experiments to visualize the consequences of the budget allocation problem and the optimization effect of our method.
> ****
> **About the experimental design:**
> The experimental design of this paper is driven by the purpose of a **fair comparison** with baselines [1,2,3]. The experimental section uses essentially the *same experimental design as baselines* (including the datasets, victim model, and attack budget) to prove our point. In addition, the analysis and visualization experiments are also designed to provide support for our point.
> ****
> **Q1: The scope of the studied attack might be too limited.**
> **A1**: There are articles that focus on the problem of untargeted gray-box poisoning attacks on the graph structure [1,2,4]. A gray-box attack represents an attacker having access to training samples of the victim model; a poisoning attack represents the victim model being retrained after data contamination. Thus, the gray-box poisoning attack is concerned with the attack transferability. This means that researchers need to mine the graph data for important as well as vulnerable input dimensions. The problem of transferability is more studied in computer vision [5-8] but less studied in graphs. That is why we consider the transferability of attack on graphs is an urgent issue to be explored.
> ****
> **Q2: About the feasibility on large scale datasets.**
> **A2**: The application of gradient-based attack methods on large scale datasets is limited by the hardware. The adjacency matrix for large datasets has a huge size taking up large amounts of memory. The computation of the gradient requires the forward process of the model to use a dense tensor as input, which results in a very large memory footprint for the gradient matrix. The memory of existing servers cannot support the implementation of the methods in this paper on large datasets.
> ****
> **Q3: Confusion caused by ‘a fixed random seed’.**
> **A3**: The initial parameters of the victim models are different. We apology for the wrong expression 'fixed random seed'. Initializing victim model for 10 times in *one python script* under a fixed random seed will have different initial parameters. It belongs to the coding stuff that should not be detailed in the main body.
> ****
> **Q4: Performance of our method on the PubMed dataset.**
> **A4**: We test GraD on PubMed (19717 nodes, 44338 edges) with the victim model being GCN. The result of each attack methods is shown as the classification accuracy of the poisoned data: Clean: 85.9%; DICE: 84.2%; EpoAtk: 83.8%; Meta_Train: 81.1%; Meta_Self: 78.4%; GraD(ours): 70.4%. We will put these results in the Appendix after the complete experiment.
> ****
> **Q5: About undefined symbols and typos.**
> **A5**: We have revised these issues in the revised version. Changes are remarked in red.

---

> > ### Author Response · Authors · 2022-08-02
> > **Reference list for authors response**
> >
> > [1] Zügner, Daniel, and Stephan Günnemann. "Adversarial attacks on graph neural networks via meta learning." arXiv preprint arXiv:1902.08412 (2019).
> > [2] Lin, Xixun, et al. "Exploratory adversarial attacks on graph neural networks." 2020 IEEE International Conference on Data Mining (ICDM). IEEE, 2020.
> > [3] Waniek, Marcin, et al. "Hiding individuals and communities in a social network." Nature Human Behaviour 2.2 (2018): 139-147.
> > [4] Liu, Zihan, et al. "Surrogate Representation Learning with Isometric Mapping for Gray-box Graph Adversarial Attacks." Proceedings of the Fifteenth ACM International Conference on Web Search and Data Mining. 2022.
> > [5] Inkawhich, Nathan, et al. "Perturbing across the feature hierarchy to improve standard and strict blackbox attack transferability." Advances in Neural Information Processing Systems 33 (2020): 20791-20801.
> > [6] Wang, Xiaosen, and Kun He. "Enhancing the transferability of adversarial attacks through variance tuning." Proceedings of the IEEE/CVF Conference on Computer Vision and Pattern Recognition. 2021.
> > [7] Demontis, Ambra, et al. "Why do adversarial attacks transfer? explaining transferability of evasion and poisoning attacks." 28th USENIX security symposium (USENIX security 19). 2019.
> > [8] Suciu, Octavian, et al. "When does machine learning {FAIL}? generalized transferability for evasion and poisoning attacks." 27th USENIX Security Symposium (USENIX Security 18). 2018.

---

> > ### Comment · Reviewer_CLc8 · 2022-08-04
> > **Reply to authors**
> >
> > I thank the authors for the reply. However, it adds up the confusion about the *motivation* of this work:
> >
> > When listing the contributions, the authors target at budget allocation. While in the answer to Q1, the authors consider the transferability as the urgent issue in their study. Then
> > - What is the main problem this paper aims to address?
> > - If it is budget allocation, then is it really reasonable to *only* consider a scenario (untargeted attack, small scale graphs) that has relatively *sufficient* budgets, instead of the scenario (targeted attack, large scale graphs) that has relatively *insufficient* budgets?
> > - If it is about transferability, how does the discovered issue relate to the transferability? To my knowledge, transferability has already been explored in the literature, such as TDGIA (Zou et al., KDD 2021), where they use plentiful defense/target models, while the authors only study three.
> >
> > Regarding the experiments, the authors aim to verify the existence of the budget allocation problem. However, without more results from different perturbation rates defense/target models, it is hard to convince the readers, specifically, whether the improvements are traded from the threats to certain defense/target models? Moreover, to my knowledge, several attack methods in other settings (please refer to my Q1 for more details) also adopt the same loss studied in the paper, but the authors haven't discussed nor analyzed.
> >
> > The random seed is quite important. I am not convinced by "Initializing victim model for 10 times in one python script under a fixed random seed will have different initial parameters. ". It's quite different from the common practice. Essentially, the values of seeds could affect several factors involved with randomness, such as the data loaders. This would add up my concerns about the reliability of the experimental results.

---

> > > ### Author Response · Authors · 2022-08-04
> > > **Response to reviewer's reply**
> > >
> > > Thank you for your reply! We have organized the responses into the following topics.
> > > ****
> > > **About motivation:**
> > > Our motivation is based on the fact that the CE loss leads to a tendency - nodes with lower confidence on the label class contribute more on the structural gradient. This phenomenon leads to the result that the attacker almost ignores the potential vulnerability of nodes with other confidence levels. The losses mentioned by the reviewers is based on ‘margin’, which filter some nodes from the overall loss without change the form of CE (the ‘margin’ method is similar to self-paced learning). Besides, those papers consider that attackers should focus on nodes near the decision boundary, as we assumed before we start our work. The discovery of this paper is a counter-commonsense phenomenon that the attacker should focus on nodes at all confidence levels equally rather than only on nodes at low confidence levels (i.e., CE losses are not good for all classification-related tasks).
> > > ****
> > > **About the scenario:**
> > > The budget in the targeted attack problem is sufficient and the number of target nodes are small, so there is no need to consider the budget allocation problem in targeted attack. Our method can theoretically be applied for small or large graphs. Our experiments in PubMed (Q4&A4 in the first response) show that the performance of our method is not affected by the increasing scale of the graph. However, as we elaborated in Q2, the hardware is not sufficient to support our method on large graphs (because sparse matrices do not support gradient backpropagation). About '*only* consider a scenario', the studying of a specific scenario is quite common. In the surveys we have previously cited, you can actually find about half of the work focusing on a certain scenario.
> > > ****
> > > **About transferability:**
> > > In fact, gray-box attacks are the study of transferable attacks. This is because we use a the surrogate model (a GCN) to generate an attack strategy to attack GNNs of unknown architecture. The transferability of attacks is a broad concept, but not often talked about in the graph domain.
> > > ****
> > > **About defense/target model:**
> > > I still insist that our experiments are consistent with the experimental setup of the compared baseline attackers, for a fair comparison, as these methods are studying the same scenario. If you read all the attack defense articles in the field, then you can find a very large number of experimental setups. Experiments exist in every article that have not been done in other articles. My understanding of the field is that the purpose of the attack model is to disrupt the generic GNN models, while the purpose of the defense model is to detect and defend against the attacks from the attack model.
> > > ****
> > > **About random seed:**
> > > If you reset the seeds once in a python file, then every time you call random numbers (e.g., initial parameters for GNN) after resetting you will get random numbers one after the other from a fixed random number sequence. Unless you reset the seed once before each initialization of the GNN model parameters, then you will get a different initialization. For example, we have a random number sequence R={r_1,r_2,...r_n}, and the GNN has k parameters. For the first time, the first GNN gets rundom number {r_1,...,r_k} for its initialization. Then, the second GNN gets {r_k+1,...,r_2k} for its initialization. This is also same for other GNNs. If you reset the seed before you initialize, for example the second GNN, then it will get {r_1,...,r_k}, which will be same as the first GNN's parameters. The code we share in the supplementary contains test.py. If you are interested in this coding trick, you can use this file to run GCN 10 times to see if the results differ.
> > > ****
> > > Our response may be a bit long. We appreciate your patience if you read this all. Please get back to us if you have any other questions and we would love to continue the discussion.
> > >
> > > Authors

---

> > > > ### Comment · Reviewer_CLc8 · 2022-08-04
> > > > **Reply to authors**
> > > >
> > > > I thank the authors for the follow-up discussion. However, if the authors don't provide further evidence to support their claims (made in the previous two responses), it's hard to convince the readers. Here I just list some of those claims in the last response:
> > > >
> > > > > This phenomenon leads to the result that the attacker almost ignores the potential vulnerability of nodes with other confidence levels.
> > > >
> > > > Why? To my understanding, the authors only analyze the most ideal case, where there is only one node and one class. Throughout the paper, I didn't see any other theoretical conclusions/proofs.
> > > >
> > > > > The losses mentioned by the reviewers is based on ‘margin’...Besides, those papers consider that attackers...
> > > >
> > > > Please give specific references to these "losses".
> > > >
> > > > > The discovery of this paper is a counter-commonsense phenomenon that the attacker should focus on nodes at all confidence levels equally rather than only on nodes at low confidence levels (i.e., CE losses are not good for all classification-related tasks).
> > > >
> > > > I agree that the discovery in this paper is interesting. However, the results seem to only be limited to an ideal scenario (theoretically) and a specific attack (empirically), since the authors didn't provide further analysis. Furthermore, "CE losses are not good for all classification-related tasks" seems to be an overclaim. Can the authors provide more justifications beyond the toy example analyzed in the paper? (Besides, all of the notations in the paper could be improved. In the review, I only list a few. However, the authors could check with all of the superscripts and subscripts, e.g., $y_i$ and $y_k$.)
> > > >
> > > > After reading the authors' response about the motivation, it seems they didn't answer my question. What is the main problem this paper aims to address? To my understanding of the authors' meaning, it's to make the budget allocation better in the attacks that adopt the CE loss. If so, since the authors didn't provide more theoretical justifications than the toy example, the readers would like to see more empirical support, which is about the experiments.
> > > >
> > > > > The budget in the targeted attack problem is sufficient and the number of target nodes are small, so there is no need to consider the budget allocation problem in targeted attack.
> > > >
> > > > If this paper aims for a more reasonable budget allocation in the attacks, then shouldn't this paper provide more experimental results with different budgets and when the budget is insufficient to better support the claims? If the authors claim that all of the attacks using CE loss would suffer from this issue, shouldn't more empirical or theoretical supports need to be provided?
> > > >
> > > > > Transferability & I still insist that our experiments are consistent with the experimental setup of the compared baseline attackers, for a fair comparison, as these methods are studying the same scenario. If you read all the attack defense articles in the field, then you can find a very large number of experimental setups. Experiments exist in every article that have not been done in other articles.
> > > >
> > > > Could the authors provide more evidential support? All of the baselines compared in the paper are from at least two years ago when few defense methods were proposed at that time. As mentioned in the review and the referred surveys, many advanced defense methods have been developed during the two years. If the gain of the proposed method in this paper is traded from the threats to other defense/target models, it may weaken (otherwise strengthen) the significance of this paper.
> > > >
> > > >
> > > > > random seed
> > > >
> > > > The devil is in the details. Especially for scientific research at a top conference, the authors could be more serious and make the clarity of the paper better.

---

> > > > > ### Author Response · Authors · 2022-08-06
> > > > > **Authors response**
> > > > >
> > > > > Thank you for your reply! Based on your response, we find that you have a big misunderstanding of the scenario covered in the methods section of this paper. We hope our reply will help you understand the methodology better.
> > > > > ****
> > > > > Q1: To my understanding, the authors only analyze the most ideal case, where there is only one node and one class.
> > > > > A1: This paper never claimed that it discussed any ideal scenario, which is ‘a graph with one node and one class’. We try to sort out your possible misconceptions.
> > > > > 1. Please note Figure 2 and Equation 6. The structural gradient A^{grad} is the gradient over the adjacent matrix generated by the loss function of all test nodes. Where the gradient generated by each node v_i is the partial gradient matrix A^{grad}_{vi}. Equation 6 shows that the average of the partial gradient matrices of the test nodes is equal to the structural gradient A^{grad}. This means that there is not just 'one node' in the scenario. We are analyzing the contribution of each node to the structural gradient in the whole structural gradient in relation to each node’s confidence level.
> > > > > 2. The formula for CE loss is mentioned in Equation 9. We guess this formula makes you think we are considering only one class. First of all, we would like to remind you that the label is a one-hot distribution, which means that only the labelled class has a value of 1, while the other classes have a value of 0. Another thing that needs to be reminded of is that the purpose of the attack is to make the model's prediction of the node deviate from the original prediction (i.e., deviate from the pseudo-label class).
> > > > > The meaning of the P_{v_i}(y_i) term in Equation 9 is the confidence level of the model's prediction on the pseudo-label class y_i. For example, if a classification task has 3 classes and the model predicts [0.1,0.2,0.7] (10% class 1; 20% class 2; 70% class 3) for a node v_k, and its pseudo-label y_k is the 3rd class, then the confidence level P_{v_i}(y_i)=0.7. We hope this makes you understand that there is more than one class in the scenario.
> > > > > 3. We visualized the contribution of all nodes to the structural gradient in Figure 3. The horizontal coordinate is the confidence level of each node on the predicted class, and the vertical coordinate is the L2 norm of the node’s partial gradient matrix. This graph illustrates that nodes with lower confidence levels tend to produce more significant partial gradient matrices and thus contribute more to the structural gradient A^{grad}. Figure 3 may be able to help you understand this paper better.
> > > > >
> > > > > If our response does not help you understand better, then you can describe more about the 'ideal scenario' you mentioned (including which part of the description made you think we are talking about the 'ideal scenario') so that we can figure out the misunderstanding.
> > > > > ****
> > > > > Q2: Please give specific references to these "losses".
> > > > > A2: [1,2] include the losses I mentioned.
> > > > > ****
> > > > > Q3: About some experiments you mentioned
> > > > > A3: As we have repeatedly emphasized, many widely cited works in the field, such as [1] and [3], employ only 5% as the budget. Our baseline [4] also employs 3% as the budget for the attack, so we include this part of the experiment as well. 'Budget allocation' is our way of explaining the problem and motivation, the essence of which is that we find that the contribution of nodes to the structural gradient is weighted by their confidence level. A visualization of this phenomenon can be found in Section 5.3. We believe that a higher attack budget setting would destroy the imperceptibility of the attack, but we still add some experiments to address your doubts.
> > > > > We extend the comparison with the baseline method Meta-Self at budgets at 10% and 20% on Cora Dataset. All the perturbed graphs involved in the following experiment can be generated by the code we provided in the supplementary.
> > > > >
> > > > > When budget=0.1*edges:
> > > > > | Victim    | GraD  | Meta-Self |
> > > > > |-----------|-------|-----------|
> > > > > | GCN       | 58.6% | 63.9%     |
> > > > > | GraphSage | 65.1% | 66.0%     |
> > > > >
> > > > > When budget=0.2*edges:
> > > > > | Victim    | GraD  | Meta-Self |
> > > > > |-----------|-------|-----------|
> > > > > | GCN       | 32.8% | 41.0%     |
> > > > > | GraphSage | 51.9% | 55.3%     |
> > > > >
> > > > > Besides, we attack the defense model RGCN [5] (pytorch implementation from DeepRobust [6]). The results:
> > > > >
> > > > > | Budget | GraD  | Meta-Self |
> > > > > |--------|-------|-----------|
> > > > > | 0.05   | 68.5% | 69.5%     |
> > > > > |   0.1  | 60.7% | 63.8%     |
> > > > > | 0.2    | 45.1% | 49.1%     |
> > > > >
> > > > > You seem to think that target model and victim model are different, but as far as we understand they have the same meaning. We hope that the results we provided will resolve the your confusion about experiments.

---

> > > > > > ### Author Response · Authors · 2022-08-06
> > > > > > **References for the response**
> > > > > >
> > > > > > [1] Xu, Kaidi, et al. "Topology attack and defense for graph neural networks: An optimization perspective." 28th International Joint Conference on Artificial Intelligence, IJCAI 2019. International Joint Conferences on Artificial Intelligence, 2019.
> > > > > > [2] Geisler, Simon, et al. "Robustness of graph neural networks at scale." Advances in Neural Information Processing Systems 34 (2021): 7637-7649.
> > > > > > [3] Zügner, Daniel, and Stephan Günnemann. "Adversarial attacks on graph neural networks via meta learning." arXiv preprint arXiv:1902.08412 (2019).
> > > > > > [4] Lin, Xixun, et al. "Exploratory adversarial attacks on graph neural networks." 2020 IEEE International Conference on Data Mining (ICDM). IEEE, 2020.
> > > > > > [5] Zhu, Dingyuan, et al. "Robust graph convolutional networks against adversarial attacks." Proceedings of the 25th ACM SIGKDD international conference on knowledge discovery & data mining. 2019.
> > > > > > [6] Li, Yaxin, et al. "Deeprobust: A pytorch library for adversarial attacks and defenses." arXiv preprint arXiv:2005.06149 (2020).

---

> > > > > > ### Comment · Reviewer_CLc8 · 2022-08-08
> > > > > > **Reply to authors**
> > > > > >
> > > > > > I thank the authors for the follow-up reply.
> > > > > >
> > > > > > > We are analyzing the contribution of each node to the structural gradient in the whole structural gradient in relation to each node’s confidence level.
> > > > > >
> > > > > > From the authors' response, it's clear that the authors understand that the final structural gradient matrix is composed of all the considered target nodes by averaging. However, all of the main discoveries in this paper seem to be derived by analyzing the contribution of a specific node from a specific class. In other words, the paper didn't draw any theoretical conclusions concerning more general cases outside the focus of the contribution from a single node to support the main claims.
> > > > > >
> > > > > > > As we have repeatedly emphasized, many widely cited works in the field, such as [1] and [3], employ only 5% as the budget. Our baseline [4] also employs 3% as the budget for the attack, so we include this part of the experiment as well.
> > > > > >
> > > > > > (I'll use the authors' references.) First, as I mentioned, All of the baselines compared in the paper are from at least two years ago, including [1,3] when few defense methods were proposed at that time. As mentioned in the review and the referred surveys, many advanced defense methods have been developed during the two years. If the gain of the proposed method in this paper is traded from the threats to other defense/target models, it may weaken (otherwise strengthen) the significance of this paper.
> > > > > >
> > > > > > Second, even those papers mentioned by the authors [1,3] conduct experiments at several perturbation rates and evaluate the transferability with several representative GNN methods. For example, the meta attack [1] from ICLR2020 (3 years ago) evaluates the 1%, 5%, 10%, 15%, 20% perturbation rates, and 3 representative GNN methods, which is far more than those conducted in the paper.
> > > > > >
> > > > > > > [1,2] include the losses I mentioned.
> > > > > >
> > > > > > This response seems to be confusing. Did I mention any of them in the review?
> > > > > >
> > > > > > Overall, I also like the interesting discovery in the paper, but the authors could develop more in-depth theoretical and empirical understandings of the discovery, and make the presentation clearer and easier for the readers to follow. I believe the paper would make a high impact on the community if the authors provide more in-depth theoretical and empirical insights behind the discovery.

---

> > > > > > > ### Author Response · Authors · 2022-08-08
> > > > > > > **Authors response**
> > > > > > >
> > > > > > > We focus on explaining the reviewers' doubts about the methodology in this response.
> > > > > > >
> > > > > > > In our view, the reviewer's main concern is that our theoretical analysis scenario and the practical application scenario are inconsistent. This is a misconception. As you said, the final structural gradient matrix is composed of all the considered target nodes by averaging. Yes it is, and we are discussing the difference between target nodes in contributing to the final structural gradient matrix. we elaborated that all partial gradient matrices from target nodes are actually weighted by a weight associated with the confidence level. This leads to the fact that each node contributes differently to the final structural gradient. When we discuss the difference between the nodes, there is no way to consider all nodes as a whole.  What we need to do is investigate to what extent each node is weighted.
> > > > > > >
> > > > > > > For example, there are 3 target nodes v_1,v_2, and v_3 with confidence level 0.1,0.5,1 at their labeled classes.  Then according to our analysis, the partial gradient matrices of v_1,v_2, and v_3 are weighted by 10, 2, 1, respectively. This makes the partial gradient matrix from v_1 dominant the final structural gradient matrix (because the final one is the average of the partial). The paper discusses that this phenomenon causes the attacker to focus on v_1 (because choosing the edge perturbation is based on the final structural gradient) and easily ignore the potential possibility of v_2 and v_3 being attacked.
> > > > > > >
> > > > > > > Hope this will help you understand the methodology part better.

---

> > > > > > > > ### Comment · Reviewer_CLc8 · 2022-08-08
> > > > > > > > **Reply to authors**
> > > > > > > >
> > > > > > > > I thank the authors for the explanation.
> > > > > > > >
> > > > > > > > To make my doubts about the gap between theoretical analysis and experimental design in the paper clearer, I'd like to highlight the following:
> > > > > > > >
> > > > > > > > In theory, the authors only analyze the contribution of a node to the final structural gradient matrix (under a simplified setup, i.e., linearized GNN, w/o considering $\partial P_{v_i}/\partial A$). However, no further implications of the biased/unbiased weighted gradients are discussed. To be more specific,
> > > > > > > >
> > > > > > > > - How would it influence the other nodes from the same/different classes as $v_i$?
> > > > > > > > - What is the difference comparing the attacks with debiased gradients to those with the undebiased/original gradients?
> > > > > > > > - Are the performance gain observed in the attacks to normal GNNs traded from attack performances to the robust GNNs?
> > > > > > > >
> > > > > > > > There are tools available in the literature for the authors to investigate more about the influences to the neighbors, e.g., [1], [2] and [3] (The last two were already available 1 year ago).
> > > > > > > >
> > > > > > > > [1] Keyulu Xu, Chengtao Li, Yonglong Tian, Tomohiro Sonobe, Ken-ichi Kawarabayashi, Stefanie Jegelka. Representation Learning on Graphs with Jumping Knowledge Networks. ICML 2018.
> > > > > > > >
> > > > > > > > [2] Yongqiang Chen, Han Yang, Yonggang Zhang, Kaili Ma, Tongliang Liu, Bo Han, James Cheng. Understanding and Improving Graph Injection Attack by Promoting Unnoticeability. ICLR 2022.
> > > > > > > >
> > > > > > > > [3] Jiong Zhu, Junchen Jin, Donald Loveland, Michael T Schaub, Danai Koutra. How does Heterophily Impact the Robustness of Graph Neural Networks? Theoretical Connections and Practical Implications. KDD ’22.

---

> > > > > > > > > ### Author Response · Authors · 2022-08-08
> > > > > > > > > **Authors response**
> > > > > > > > >
> > > > > > > > > Thank you for your reply.
> > > > > > > > >
> > > > > > > > > Before answering the three questions specified by the reviewer, we would like to do some clarification.
> > > > > > > > > First, a simplified setup, i.e., linearized GCN, is an empirically better-performing surrogate model. This is not the subject of this paper, but linearized GCNs do perform better than nonlinear GNNs as surrogate models. It is not saying that we consider the linearized GCN as an example victim model. The surrogate model and the victim model are different and do not need to be the same. Our aim is to obtain a transferable attack strategy for the victim model using the gradient information from the surrogate model.
> > > > > > > > > Second, the partial gradient matrix is actually present in all gradient-based methods. However, those methods consider the final structural gradient as a whole rather than discussing the partial ones independently.
> > > > > > > > > Figure 3 (in Section 5.3) verifies the relationship between the partial gradient matrices of all test nodes and their confidence. This figure may be placed in the methodology section or referenced in the methodology so that it will help audiences to understand better.
> > > > > > > > > ****
> > > > > > > > > Q1: How would it influence the other nodes from the same/different classes as vi?
> > > > > > > > > A1:  The ‘influence’ depends on whether the nodes are connected rather than whether the nodes are in same/different classes. The features of node v_i are passed to its neighbors in the process of aggregation (message passing). For a k-layer GCN, the features of node v_i will be passed to its neighbors within k-hop. Thus, the loss of node v_i is able to generate gradients (via backpropagation) on the features of these neighbors, and on the edges between node v_i and these neighbors.
> > > > > > > > > ****
> > > > > > > > > Q2: What is the difference comparing the attacks with debiased gradients to those with the undebiased/original gradients?
> > > > > > > > > A2: We actually discussed this issue in both the Introduction and Method, but we didn't explicitly name 'original' and 'debiased' gradient. Regarding original gradients, since nodes with low confidence levels tend to produce larger partial gradient on the graph structure, the attacker using original structural gradient is more likely to *perturb the edges involved in the message passing of low confidence nodes*. This leads to problems such as 1. attackers always try to make the confidence level of low confidence nodes lower, and thus the contribution of low confidence nodes to the final gradient becomes larger. 2. attackers tend to ignore nodes of relatively higher confidence levels, although they may be vulnerable (since aggregation will forcibly fuse the features of two connected nodes). Debiased gradients allow the attacker to better focus on the global vulnerable nodes compared to original gradients.
> > > > > > > > > ****
> > > > > > > > > Q3: Are the performance gain observed in the attacks to normal GNNs traded from attack performances to the robust GNNs?
> > > > > > > > > A3: In our previous response, we added a set of experiments on the defense(robust) model RGCN. We chose RGCN because it is open-sourced in DeepRobust. We can reproduce the RGCN and show the results to the reviewers during the discussion session. The previously shown results demonstrate the effectiveness of our model for RGCN.
> > > > > > > > > ****
> > > > > > > > > We have recently read [2] mentioned by the reviewer. It's a interesting work that argues the tendency of node injection in adding edges between nodes with very different attributes. [3] seems to be published recently. It is interesting to relate edge perturbations to the change in homophily. We did not notice [2,3] before completing this article, as they were not published. We will discuss them in related work.

---

> > > > > > > > > > ### Comment · Reviewer_CLc8 · 2022-08-09
> > > > > > > > > > **Comment on author response**
> > > > > > > > > >
> > > > > > > > > > I thank the authors for the follow-up explanation.
> > > > > > > > > >
> > > > > > > > > > From the response of A1 to A2, the authors seem not to be aware of the graph's connectivity during the adversarial attack. By "How would it influence the other nodes from the same/different classes as $v_i$?", it refers to that, by adding the contribution of $A^{\text{grad}}_{v_i}$ to $A^{\text{grad}}$, it is expected to increase the loss of node $v_i$ by gradient ascend after the perturbation, Yes. But furthermore,
> > > > > > > > > > - What is its influence to the neighbors of $v_i$ from testset/targetset that have the same or different labels with $v_i$?
> > > > > > > > > > - How would the influences be changed by debiasing $A^{\text{grad}}_{v_i}$?
> > > > > > > > > >
> > > > > > > > > > The reason for considering the neighbors is because all of the target nodes will be considered when calculating the attack performance and these nodes could appear in the neighbors of $v_i$ and be influenced by the debiasing. I couldn't find any corresponding discussions in the paper. Hence it's unclear theoretical implications of the debiasing to other target nodes, and similar to the overall attack performances in theory. Therefore, as I asked in previous replies and my initial review, that the authors should provide more empirical analysis, yet the relevant experiments are severely lacking.
> > > > > > > > > >
> > > > > > > > > > Regarding A3, the authors are aware that "Our aim is to obtain a transferable attack strategy for the victim model using the gradient information from the surrogate model.", yet the experiments seem to attack the RGCN directly. Moreover, more perturbation rates and more defense models are needed to fully justify the improvements of the method (which has been already and repeatedly asked since my initial review).
> > > > > > > > > >
> > > > > > > > > > By the way, both [2,3] have already been available since last year. [2] is accessible via openreview since Oct. 2021. For [3], please check out the arxiv number: https://arxiv.org/abs/2106.07767 .
> > > > > > > > > >
> > > > > > > > > > Overall, I'd also like to note that I also like the interesting discovery in the paper, and I believe **the paper would make a high impact to the community if the authors provide could provide in-depth theoretical and empirical insights behind the discovery.**

---

> > > > > > > > > > > ### Author Response · Authors · 2022-08-09
> > > > > > > > > > > **Follow-up response (1)**
> > > > > > > > > > >
> > > > > > > > > > > Thanks to the reviewer for the reply.
> > > > > > > > > > >
> > > > > > > > > > > This may be our last response before the end of the discussion session. We gradually start to realize the reviewer's misunderstanding of the methodology and we try to help reviewers understand it using a simple example (it can be really helpful).
> > > > > > > > > > > ****
> > > > > > > > > > > **About [2,3]:** We have found and read [3] and cited it in the paper, since it has been admitted by the KDD committee and we also recognize the significance of this work. We rarely refer to unpublished studies because they are not yet recognized by high-level conferences. In addition, they are often not open-sourced before they are accepted. As you know, there are lots of papers with irreproducible experimental results (some of them even provide fake results without open-sourced code), so we are used to not refer to arxiv.
> > > > > > > > > > > ****
> > > > > > > > > > > **Misunderstanding: ‘yet the experiments seem to attack the RGCN directly.’**
> > > > > > > > > > > Our attack is transferred from a linear GCN (the surrogate model) to unknown GNNs (victim models). We are not attacking RGCNs directly. What the reviewers consider as directly attacks on RGCNs belong to *white-box attacks* rather than gray-box attacks. The difference between white-box and gray-box attacks includes if the attacker can get gradient from the victim model.
> > > > > > > > > > > **Regarding to the experiments:** We have expanded the perturbation rate to 10% and 20% and have provided the results in the previous response. We think this concern has been well-solved. Every experiment and visualization we provide in the experimental section is very important (especially the visualization) and they are sufficient to support the methodology. The experiments on more defense models than RGCN suggested by the reviewers may be expanded in the appendix rather than in the main body. We will try to reproduce some open-sourced defense models. If time allows, the complete results on multiple defense models may be expanded as a table in the Appendix.
> > > > > > > > > > > ****
> > > > > > > > > > > **Misunderstanding (the most crucial): the reviewer has not yet understood this paper in the right perspective.**
> > > > > > > > > > >
> > > > > > > > > > > 1. **What is its influence to the neighbors of vi from testset/targetset that have the same or different labels with vi?**
> > > > > > > > > > >
> > > > > > > > > > > The reviewer's understanding of the method is biased. It is not an issue of how the loss of increasing v_i will affect its neighboring nodes. Reviewer should understand the structural gradient and this paper in terms of 'how changing an edge will affect its associated nodes'. Let us take the simplest example to try to make the reviewer understand. Suppose a scenario with three nodes **v_1**,**v_2** and **v_3** and two edges **e_1** and **e_2**.
> > > > > > > > > > >
> > > > > > > > > > > The gradients generated by node *v_1* through attack loss (assume it is the undebiased loss) are **0.5** and **0.3** on *e_1* and *e_2*, respectively; the gradients generated by node *v_2* through attack loss are **0.3** and **0.1** on *e_1* and *e_2*, respectively; the gradients generated by node *v_3* through attack loss are **-0.2** and **0.1** on *e_1* and *e_2*, respectively.
> > > > > > > > > > >
> > > > > > > > > > > From a global perspective, the three nodes produce a total gradient of **0.6** and **0.5** on *e_1* and *e_2* (‘total’ or ‘average’ do not affect the comparison of *e_1* and *e_2*. ‘total’ is easier to recalculate by the reviewer). This means that changing the state of edge *e_1* is more likely to have a larger effect on the global performance (since the attack loss produces a more significant gradient on *e_1*).
> > > > > > > > > > >
> > > > > > > > > > > The above scenario is just an example to better help the reviewer understand. In the real case, we also need to consider the state of the edge (0 or 1) and whether the gradient is positive or negative (details in Equation 7).
> > > > > > > > > > >
> > > > > > > > > > > **Follow-up by response (2)...**

---

> > > > > > > > > > > > ### Author Response · Authors · 2022-08-09
> > > > > > > > > > > > **Follow-up response (2)**
> > > > > > > > > > > >
> > > > > > > > > > > > 2. **How would the influences be changed by debiasing A_{vi}^{grad}?**
> > > > > > > > > > > >
> > > > > > > > > > > > In the above example, we did not mention the confidence levels of **v_1**, **v_2** and **v_3**. We continue to assume that they have **confidence levels** of **0.1**, **0.5**, and **0.5**, respectively. Cross-entropy loss is suitable for classification tasks because it 'amplifies' the loss generated by low confidence nodes (which allows the classification model to preferentially fit the misclassified nodes). This is because the derivative of the cross-entropy function with respect to the node confidence **P** is P^{-1} (Equation 9-11). Different from classification tasks, the goal of this paper is to mislead the predicted class of test nodes. **Low-confidence nodes similar to v_1 are already more likely to be misleading**, so ''amplifying' the losses of nodes like *v_1* yield a waste of attack budget. Our proposed approach aims to solve this problem.
> > > > > > > > > > > >
> > > > > > > > > > > > Let’s get back to the example scenario. On edges **e_1** and **e_2**, the gradients of node *v_1* before it is ‘amplified’ (i.e., after debiased) are **0.05** and **0.03** ((0.5,0.3) × 0.1); the gradients of node *v_2* before it is ‘amplified’ are **0.15* and *0.05** ((0.3,0.1) × 0.5); the gradients of node *v_3* before it is ‘amplified’ are **-0.1** and **0.05** ((-0.2,0.1) × 0.5). From a global perspective, the total gradients generated by all three nodes on *e_1* and *e_2* are **0.1** and **0.13**. This means that changing the state of *e_2* is more likely to affect the model predictions globally.
> > > > > > > > > > > >
> > > > > > > > > > > > Does this example help you understand the difference between the debiased one and the undebiased one?

---

> > > > > > > > > > > > > ### Comment · Reviewer_CLc8 · 2022-08-09
> > > > > > > > > > > > > **Comment on author replies**
> > > > > > > > > > > > >
> > > > > > > > > > > > > Can the authors consider the influences of the gradients to broader nodes? The network contains more than 3 nodes. If the adversary follows the gradient direction pointed by the gradient from $v_i$, will it also affect the losses on neighbors of $v_i$ that have the same class as $v_i$? If so, how can the authors term such behavior a "waste", when without any evidence?

---

> > > > > > > > > > > > > > ### Author Response · Authors · 2022-08-09
> > > > > > > > > > > > > > **Authors response**
> > > > > > > > > > > > > >
> > > > > > > > > > > > > > We thank the reviewer for the replys.
> > > > > > > > > > > > > >
> > > > > > > > > > > > > > Reviewer may be more familiar with node injection attacks, as reviewer is concerned with pollution propagation through edges to a test node and its neighboring nodes.
> > > > > > > > > > > > > > Due to the aggregation mechanism of GNN, changing one edge e_ij certainly affects more than one node (v_i, v_j and their neighboring nodes). However, based on the original/undebiased gradient, the attacker is likely to select an edge mainly because a node with low confidence level contributes a very significant gradient value on one of its nearby edges (edge state can be 0 or 1). 'Waste' does not mean that an edge perturbation is not able to reduce the performance of the victim model, but that the attacker could have a more reasonable strategy leading to better attack performance. The visualizations in Figure 3 and 4 are the empirical evidence. The methodology in the paper is talking about broader nodes. There are only three nodes in the example just making the example easy to understand.
> > > > > > > > > > > > > >
> > > > > > > > > > > > > > We have acknowledged the value of [3] in our response, rather than considering it as an unreliable arxiv. The reviewers felt that we avoided testing on defense models. In the limited discussion period, we have provided some experiment on larger attack budget, RGCN and the white-box setting in the response to reviewer 2pFy. Reproducing the work of others takes time, and authors need to prepare other submissions before the near deadline. We have mentioned in our response that we plan to reproduce some other open-source defense models for testing. Does the reviewer suggest us to test on the defense model of [3]? If so, we will prioritize this job. If not, the reviewer can feel free to recommend specified reproducible defense models.
> > > > > > > > > > > > > >
> > > > > > > > > > > > > > Gray-box attack setting has the clear definition that the victim model should be unknown, which is mentioned in the surveys cited in the review. We were not trying to dispute the reviewer's knowledge in this subfield, but to clarify the experimental scenario. Since white-box attacks, unlike gray-box settings, do not qualify for the study of transferable attacks, we spent words on clarification.

---

> > > > > > > > > > > > > > > ### Comment · Reviewer_CLc8 · 2022-08-10
> > > > > > > > > > > > > > > **Comment on author reply**
> > > > > > > > > > > > > > >
> > > > > > > > > > > > > > > I thank the authors for the late reply.
> > > > > > > > > > > > > > >
> > > > > > > > > > > > > > > First, my point isn't about injection or modification, instead, I am concerned about the theoretical part in the current version of the paper. The authors just show an interesting discovery, which however isn't shown to imply any **rigorous conclusions**.
> > > > > > > > > > > > > > >
> > > > > > > > > > > > > > > No matter whether the adversary takes an injection strategy or not, the perturbed edges would influence the neighbors, is that correct? I list the recent works including both injection and modification, with the hope that the authors could use the tools to provide rigorous analysis of their interesting discoveries.
> > > > > > > > > > > > > > >
> > > > > > > > > > > > > > > However, the authors use a lot of intuitive descriptions but one can hardly justify the claims. Overall, the authors **didn't directly reply** to my two main concerns.
> > > > > > > > > > > > > > >
> > > > > > > > > > > > > > > As for the response to the theoretical part, the authors use lots of intuitive descriptions without rigorous support:
> > > > > > > > > > > > > > > > However, based on the original/undebiased gradient, the attacker is likely to select an edge mainly because a node with low confidence level contributes a very significant gradient value on one of its nearby edges (edge state can be 0 or 1).
> > > > > > > > > > > > > > >
> > > > > > > > > > > > > > > > but that the attacker could have a more reasonable strategy leading to better attack performance.
> > > > > > > > > > > > > > >
> > > > > > > > > > > > > > >
> > > > > > > > > > > > > > > The two points above are actually what I'd like to see the authors could provide formal justifications.
> > > > > > > > > > > > > > >
> > > > > > > > > > > > > > >
> > > > > > > > > > > > > > > **If without formal results**, the paper is expected to provide more empirical understandings, in other words, results at more perturbation rates and more defense models that could cover the literature, and provide more understanding of the discovery based on the empirical observations. However, the current empirical support is too limited to understanding the behavior (as I pointed out in the previous replies, less than Meta-attack that comes from ~3 years ago).
> > > > > > > > > > > > > > > **All of the concerning results are critical to evaluating the contributions and claims of the paper**. It's hard to evaluate the paper if the results are detained to the camera-ready phase. The defense models used in [2,3] have high coverage of the existing literature. However, the authors are suggested to refer to the *comprehensive collections provided by the authors of the surveys cited by the paper*.
> > > > > > > > > > > > > > >
> > > > > > > > > > > > > > >
> > > > > > > > > > > > > > > The other two minor points:
> > > > > > > > > > > > > > >
> > > > > > > > > > > > > > > I use the authors' descriptions of RGCN to exemplify many of the *unsupported claims* of the authors. The authors always begin with an accuse of misunderstanding of me but can't provide direct answers. The authors could show more respect for the efforts of all reviewers during the rebuttal/discussion, by showing the evidence and direct answers.
> > > > > > > > > > > > > > >
> > > > > > > > > > > > > > > The other claim of "fair comparison" seems to be another misconception of the authors. If we compare a baseline from many years ago, like a multi-layer perceptron for example, does "fair comparison" refer to we only need to compare the baselines in the multi-layer perceptron paper?

---

> > > > > > > > > > > > ### Comment · Reviewer_CLc8 · 2022-08-09
> > > > > > > > > > > > **Comment on the authors' reply**
> > > > > > > > > > > >
> > > > > > > > > > > > It's not me that misunderstands the authors' experiments. The authors described that "Besides, we attack the defense model RGCN [5] (pytorch implementation from DeepRobust [6]). The results:", so **HOW CAN I KNOW WHAT EXACTLY YOU ARE ATTACKING?**
> > > > > > > > > > > >
> > > > > > > > > > > > The authors could be more serious about their claims, including the claims like "misunderstanding", "misconception" in the rebuttal, and their claims in the paper.

---

> > > > > > > > > > > > ### Comment · Reviewer_CLc8 · 2022-08-09
> > > > > > > > > > > > **Comment on the authors' reply**
> > > > > > > > > > > >
> > > > > > > > > > > > About the literature review and experiments, I have already pointed out the surveys and a specific defense methods **accepted** in previous NeurIPS conference, but the authors keep avoiding providing more empirical support and finding excuses like "we are used to not refer to arxiv.". Please be more serious about YOUR RESEARCH.

---

### Official Review · Reviewer_2pFy · 2022-07-10

**Rating:** 7
**Confidence:** 3
**Soundness:** 3 good
**Presentation:** 3 good
**Contribution:** 3 good

**Summary:**

- The authors tackle the problem of the attack objective (the form of cross-entropy function) in the untargeted attacks on node-level classification models.
- They show that nodes with low confidence significantly affect the gradient.
- To alleviate this problem of inefficient attack budgets, they propose a novel attack objective whose corresponding gradient is not affected by the confidence of nodes.
- They conduct experiments on gray-box poisoning attack and shows that the proposed attack method GraD (based on the proposed gradient-debias attack objective) can significantly improve the attack performance.

**Questions:**

- I hope that the explanation of the purpose of the poisoning attack is simply added to the introduction or related works.
- Why are there no line numbers in your submission?
- I want to know more scenarios to apply GraD other than gray-box / white-box attack on GCN-based node classification models if exist.
- If I understood correctly, we can apply GraD to white-box attack too. Can you show some white-box attack results of GraD and white-box baseline methods?
- One can replace the attack objective of GraD to the original negative cross-entropy loss and compare it with GraD as an ablation study. Is there already a baseline corresponding to it in the table 1, 2? If not, I want to see this result.

POST REBUTTAL COMMENTS: The authors answer the questions. My concerns have been addressed. They can apply their method also in the white-box attack setting. They add broader impacts and fix some minor issues in the paper. I adjust my score from 6 to 7.

**Limitations:**

This paper handle the adversarial attack which can be used by malicious adversaries. That said, I wish that the authors include ethics or broader-impact statement in the paper or appendix.

**Strengths And Weaknesses:**

# Strengths
- The authors states an interesting problem in the cross-entropy based attack objective. (Figure 1 was very helpful to understand the concept!)
- The authors explain the proposed problem using a simple mathematics. Also, they mathematically explain their proposed attack objective can solve the problem.
- They conduct fair comparison on various datasets and achieve the state-of-the-art attack performance.

# Weaknesses
- The usage of the method seems limited to gray-box / white-box attack on GCN-based node classification models.
- Ablation studies seem insufficient.

---

> ### Author Response · Authors · 2022-08-02
> **Rebuttal response by authors**
>
> We are grateful to you for your effort in reviewing our paper. We will next respond to the weaknesses and questions you mentioned.
> ****
> **Q1: Is there already a baseline corresponding to it in the table 1&2?**
> **A1**: The Meta-Self [1] is the baseline you mentioned, and comparing it with our method can be considered an ablation study. Both Table 1 and 2 contain a comparison of our method with this baseline.
> ****
> **Q2: Can you show some white-box attack results of GraD and white-box baseline methods?**
> **A2**: We conduct an poisoning attack comparing with the white-box method CE-minmax [2] (implementation from DeepRobust [3]). Our proposed GraD is implemented in the CE-minmax’s backbone and test scenario. The attack budget is set to 5%. On the Cora dataset (Acc=82.3%): GraD: 74.4%, CE-minmax: 75.8%; on the Cora-ML dataset (Acc=83.1%): GraD: 74.7%, CE-minmax: 76.5%. The results demonstrate the applicability of our method to white-box attack methods.
> ****
> **Q3: I hope that the explanation of the purpose of the poisoning attack is simply added to the introduction or related works.**
> **A3**: The poisoning attack studies the impact of attacked data on the model training. It is also a scenario encountered in an attack where the attacker has contaminated the data before the model is trained. We have added this sentence to the Introduction, marked in red (line 30-32).
> ****
> **Q4: I want to know more scenarios to apply GraD other than gray-box / white-box attack on GCN-based node classification models if exist.**
> **A4**: This paper discusses the generation of the gradient on the graph structure from the node level. Thus, it is a global budget allocation for the nodes (i.e., samples). Our method has the potential to be applied to attacks on various data types that require budget allocation, such as model skewing with data poisoning. In addition, the method can be used in attribution analysis studies of node-level classification models.
> ****
> **Q5: Why are there no line numbers in your submission?**
> **A5**: Sorry for missing the line numbers. It has been fixed in the revised version.
> ****
> **Q6: I wish that the authors include ethics or broader-impact statement in the paper or appendix.**
> **A6**: Our proposed attacker is at risk of being used maliciously. A graph data holder should protect his graph structure, node attributes, and labels of training nodes. It has been added in Appendix as Broader Impact (line 476-481).
> ****
> [1] Zügner, Daniel, and Stephan Günnemann. "Adversarial attacks on graph neural networks via meta learning." arXiv preprint arXiv:1902.08412 (2019).
> [2] Xu, Kaidi, et al. "Topology attack and defense for graph neural networks: An optimization perspective." 28th International Joint Conference on Artificial Intelligence, IJCAI 2019. International Joint Conferences on Artificial Intelligence, 2019.
> [3] Li, Yaxin, et al. "Deeprobust: A pytorch library for adversarial attacks and defenses." arXiv preprint arXiv:2005.06149 (2020).

---

### Official Review · Reviewer_9RmM · 2022-07-12

**Rating:** 5
**Confidence:** 3
**Soundness:** 2 fair
**Presentation:** 4 excellent
**Contribution:** 2 fair

**Summary:**

This paper studies the problem conducting untargeted poisoning attacks against graph neural networks by pertaining edges of the original graph. The authors identified the problem of attack loss design, which mostly uses the same loss function (i.e., negative cross entropy) for model training, and pointed out that, with the wrong loss function, attack budget is wasted on causing misclassification for nodes that are already misclassified. Then a debasing solution is proposed by multiplying the gradients with their confidence scores and the obtained simple attack strategy outperforms the existing baselines significantly.

**Questions:**

I do not list this as a question that will change my decision, but I am curious to know, if there are any other ways (e.g., designing new attack loss) to perform the edge perturbation instead of renormalizing the gradient.

**Strengths And Weaknesses:**

I like the idea of the paper as it challenges some of the common beliefs in the attack design against graph neural networks, which is novel and signficant. The presentation of the paper is also very clear and easy to follow. Overall, the strength of the paper significantly outweighs the weakness. However, I have the following concerns:
1) the work by Ma et al., [9] (reference in the paper) and one missing work linked below [1] might deserve more discussion, as they are more related to the key problem identified in the paper: the downside of choosing a wrong type of attack loss.
[1] Ma et al., "Adversarial Attack on Graph Neural Networks as An Influence Maximization Problem", WSDM 2022.
2) the identified problem and fixable solution given in the paper is only demonstrated for the case of cross-entropy loss, and so the conclusion is not very generic.

---

> ### Author Response · Authors · 2022-08-02
> **Rebuttal response by authors**
>
> We are grateful to you for your effort in reviewing our paper. We will next respond to the weaknesses and questions you mentioned.
> ****
> **Q1: About papers [1,2] from Ma et al..**
> **A1**: Ma et al. [1,2] focused on black-box attacks on node attributes. These works start from the importance of nodes in the graph and introduce prior knowledge to assist the attacker in selecting the nodes to be attacked. The scenarios studied in [1,2] differ from the attack scenarios discussed in this paper (i.e., gray-box poisoning attack vis edge perturbation). We add paper [2] as a reference in the related work.
> ****
> **Q2: About how prevalent cross-entropy-based attack loss is in the field.**
> **A2**: Node classification is a common task in graph datasets. In existing white- and gray-box attacks for classification models, the form of cross-entropy is an essential component of the attack loss [3-6]. Our proposed confidence-oriented attack loss overturns this mindset, which has been justified in this paper.
> ****
> **Q3: Existence of other edge perturbation methods.**
> **A3**: In addition to edge perturbation with gradient, methods exist in the field to solve edge perturbation problems with reinforcement learning (RL), such as RL-S2V [7] and ReWatt [8] (focused on graph-level classification). Perturbation strategy optimization based on RL is an interesting research topic; however, it is not the most efficient in this paper's task because reinforcement learning is computationally expensive.
> ****
> [1] Ma, Jiaqi, Shuangrui Ding, and Qiaozhu Mei. "Towards more practical adversarial attacks on graph neural networks." Advances in neural information processing systems 33 (2020): 4756-4766.
> [2] Ma, Jiaqi, Junwei Deng, and Qiaozhu Mei. "Adversarial Attack on Graph Neural Networks as An Influence Maximization Problem." Proceedings of the Fifteenth ACM International Conference on Web Search and Data Mining. 2022.
> [3] Xu, Kaidi, et al. "Topology attack and defense for graph neural networks: An optimization perspective." 28th International Joint Conference on Artificial Intelligence, IJCAI 2019. International Joint Conferences on Artificial Intelligence, 2019.
> [4] Wu, Huijun, et al. "Adversarial examples for graph data: deep insights into attack and defense." Proceedings of the 28th International Joint Conference on Artificial Intelligence. 2019.
> [5] Zügner, Daniel, and Stephan Günnemann. "Adversarial attacks on graph neural networks via meta learning." arXiv preprint arXiv:1902.08412 (2019).
> [6] Geisler, Simon, et al. "Robustness of graph neural networks at scale." Advances in Neural Information Processing Systems 34 (2021): 7637-7649.
> [7] Dai, Hanjun, et al. "Adversarial attack on graph structured data." International conference on machine learning. PMLR, 2018.
> [8] Ma, Yao, et al. "Attacking graph convolutional networks via rewiring." arXiv preprint arXiv:1906.03750 (2019).

---

> > ### Comment · Reviewer_9RmM · 2022-08-07
> > **Thanks for the clarification**
> >
> > Thanks for the response, the raised clarification issues are well addressed by the authors. I also suggest the authors to include more discussion (mostly in your current response) regarding the previously raised concerns in the paper.

---

> > > ### Author Response · Authors · 2022-08-07
> > > **Thank you for the reply**
> > >
> > > Whether this paper is accepted or not, we will add the discussion with reviewers in the next version. Thank you for your suggestion!

---

### Author Response · Authors · 2022-08-02
**About revised version and rebuttal**

We thank the reviewers for their comments and suggestions. We have revised the paper based on some of the suggestions and marked them in red. The revisions include:
1. Add a brief intruduction for the poisoning attack. (**Line 30-32**)
2. Correct symbols and typos. (**Line 115,118,131,137,169-170,312-313**)
3. Correct a statement that could cause ambiguity. (**Line 249**)
4. Add section Broader Impact. (**Line 476-481**)

We answer to each reviewer's questions individually in our response.
****
Discussions with reviewers and some relevant literature have been added to the latest version .

---

### Meta-Review · Area_Chair_RQWp · 2022-08-24

**Recommendation:** Accept
**Confidence:** Less certain

**Metareview:**

The authors study graph modification attack (through editing the edges) in the setting of untargeted poisoning and show that negative cross entropy is not a good candidate for the attack loss. Instead they propose a novel attack objective to study the problem.

The reviewers found the topic timely and of interest to the community. They felt that the theoretical and empirical analysis could be improved to validate their claims, but overall the positives seemed to outweigh the negative perceived by the reviewers.

**Award:**

No

---

### Decision · Program_Chairs · 2022-09-14

Accept